# Antibacterial and Antioxidant Activities of Hydroalcoholic and Phenolic Extracts from *Ternstroemia dentisepala* and *T. lineata* Leaves

**DOI:** 10.3390/plants13172515

**Published:** 2024-09-07

**Authors:** Alexis Uriel Soto Díaz, María Luisa Villarreal, Marcelo Victorio-De los Santos, Alexandre Toshirrico Cardoso-Taketa

**Affiliations:** 1Centro de Investigación en Biotecnología, Universidad Autónoma del Estado de Morelos, Cuernavaca 62210, Mexico; uriel.soto.diaz@outlook.com (A.U.S.D.); luisav@uaem.mx (M.L.V.); 2Unidad Académica de Ciencias Químicas Biológicas y Farmacéuticas, Universidad Autónoma de Nayarit, Tepic 63190, Mexico

**Keywords:** *Ternstroemia lineata* DC., *Ternstroemia dentisepala* B.M. Barthol., Pentaphylacaceae, antibacterial, antioxidant, total phenolic content, TLC, HPLC, ^1^H NMR

## Abstract

Traditional Mexican medicine commonly uses infusions of *Ternstroemia* spp. to treat insomnia, injuries, and infections. The antibacterial activities of *Ternstroemia dentisepala* and *Ternstroemia lineata* were evaluated for the first time against a panel of Gram-positive and Gram-negative bacteria that have implications for human health, including *Enterococcus faecalis*, *Streptococcus agalactiae*, *Streptococcus pyogenes*, *Salmonella typhi*, *Pseudomonas aeruginosa*, and *Vibrio parahaemolyticus*. Furthermore, the scavenging potential of the hydroalcoholic (HAEs) and total phenolic extracts (TPEs) from the leaves of both plants by a 2,2’-azino-bis(3-ethylbenzothiazoline-6-sulfonic acid) assay (ABTS^•+^) was determined. Also, the total phenolic contents of the HAEs using the Folin–Ciocalteu reagent were assayed. *T. dentisepala* HAE and TPE were active against all bacterial strains tested, with a minimum inhibitory concentration between 1.0 and 6.0 mg/mL, with the last one being the most active. However, the *T. lineata* extracts only demonstrated effectiveness against *S. typhi* and *P. aeruginosa*. The TPEs from *T. dentisepala* and *T. lineata* improved the activity by approximately 30% in all bacteria tested in comparison with the HAEs. The *T. dentisepala* HAE had a higher total phenolic content than the *T. lineata* extract, which was consistent with its ABTS^•+^-scavenging activity. The two HAEs had different chemical profiles, mostly because of the types and amounts of phenolic compounds they contained. These profiles were obtained using thin-layer chromatography (TLC), high-performance liquid chromatography (HPLC), and proton nuclear magnetic resonance (^1^H NMR) experiments.

## 1. Introduction

Mexico has nine of the approximately one hundred species of the *Ternstroemia* genus (Pentaphylacaceae) reported around the world. These species are endemic or practically restricted to the country [1,2,3], and six of them are included in the IUCN Red List Threatened Species. *Ternstroemia dentisepala* B.M. Barthol is considered vulnerable, while *T. Ternstroemia lineata* DC. is of least concern. These species are known as “tila” or “trompillos” in Mexican traditional medicine [1]. The tila flower, in the form of its dried calyx, is widely marketed in medicinal plant markets and health food stores, in addition to being found in supermarkets in the form of teas, which are recommended to treat insomnia and as a tranquilizer. A review lists *Ternstroemia* spp. as one of the 24 native medicinal plants more commonly used to make herbal products in Mexico [4]. In a study of two of the most important traditional markets in the state of Hidalgo, where the Otomí culture is predominant, 20.3% of the most mentioned medicinal plants were *Ternstroemia* spp. Its flowers and fruits are used to calm the nerves and treat insomnia, as well as to treat coughs and fevers caused by infections [5]. Traditional Mexican medicine uses *Ternstroemia* spp. to relieve rheumatic pain [6]. In the west of the republic, the leaves of tila are used in the form of plasters to treat wounds, pain, bumps, and urinary infections [7].

In Mexico, the two species of *the Ternstroemia* genus with large distributions are *T. lineata* and *T. dentisepala*. Both species primarily inhabit the central area of the country, with *T. dentisepala* exhibiting a larger distribution toward the shoal and west, while *T. lineata* tends to lean more toward the south and east. Both have similar morphological characteristics, such as blade length, petiole length, and apex angle width, that make them more difficult to differentiate. Given their morphological similarities, the botanical classification of these species is frequently misunderstood. In fact, experts have reviewed and reclassified the *T. lineata* material we previously collected at Huitzilac Municipality in Morelos State, Mexico, which we formerly named *T. pringlei* [8].

The World Health Organization (WHO) has issued a warning about the urgency and need for the discovery of new antibacterial molecules in order to develop medicines to fight the growing global resistance of bacteria to antimicrobial drugs. The highly critical priority tier includes *Acinetobacter baumannii*, *Pseudomonas aeruginosa*, and various *Enterobacteriaceae*, followed by a second group of bacteria that includes *Salmonella typhi*, *Enterococcus faecium* (vancomycin-resistant), *Staphylococcus aureus*, (methicillin-resistant and vancomycin-resistant), and *Helicobacter pylori* (clarithromycin-resistant) [9].

Plants are the source of 25–50% of patented medicines, but until now, they do not yield any antibacterials used in therapeutics [10]. However, plants produce an enormous variety and complexity of secondary metabolites, many of which have antibacterial activity, leaving us eager to discover and explore new “leads”. According to reports on *T. gymanthera*, the leaf extracts showed high antimicrobial activity, indicating that they contain antimicrobial compounds. As a result, *Ternstroemia* spp. represents a potential source of new antibacterial medicines because many of their ethnomedical uses are related to bacterial infection problems [11].

Phytochemical studies of *Ternstroemia* species have reported the isolation of ursane and oleanane triterpenes [12], phenylethanoid glucosides (ternstrosides A–F), a kaempferol derivative [13], an aldehydic carotenoid [14], triterpenoid saponins [15,16], terpenoids, and aromatic compounds [17]. In a previous study, we found that the flowers of *T. lineata* contain a sedative quinone called jacaranone [8]. Furthermore, we described the radical-scavenging and antioxidant activities of the methanolic extracts from different plant organs of *T. lineata* using the ABTS^•+^ test, as well as the H_2_O_2_ protection model with *Saccharomyces cerevisiae*, where the glycosylated phenylethanoid ternstroside B was isolated and tested [18]. More recently, we evaluated the antioxidant potential of terngymnoside C and hydroxytyrosol-1-glucoside, which were isolated from the flower buds of *Ternstroemia lineata*, as well as katsumadin, which was obtained from the seedless fruits of the plant, using ABTS^•+^ and H_2_O_2_-*Saccharomyces cerevisiae* models. In silico docking analysis showed that the affinity forces of the isolated compounds to the aquaporin monomers of the modeled *S. cerevisiae* protein 3 (AQP3) and human protein 7 (AQP7) channels should regulate H_2_O_2_ cell transport, where an antioxidant mechanism based on blocking the H_2_O_2_ passage mediated by aquaporins was proposed [19]. In the current work, we investigated the antibacterial and radical-scavenging potential of the hydroalcoholic and total phenolic extracts of *T. dentisepala* and *T. lineata*. We found significant differences in these activities, which were correlated with the chemical profiles using thin-layer chromatography (TLC), high-performance liquid chromatography (HPLC), and ^1^H nuclear magnetic resonance (^1^H NMR) analyses.

## 2. Results

### 2.1. Hydroralcoholic and Total Phenolic Extracts

The hydroalcoholic extracts (HAEs) of *T. lineata* and *T. dentisepala* presented similar yields of 14.58% ± 0.63 and 16.61% ± 0.82, respectively, with the latter being slightly higher. The dry powder of both crude extracts presented a greenish-brown color. However, *T. dentisepala* produced a higher yield in terms of the total phenolic extract (TPE), with a yield of 27.5% ± 3.53, compared with *T. lineata*, which produced a yield of 15.75% ± 1.76. Both dry extracts presented a yellow-orange color, with the total phenolic extract of *T. lineata* having a paler appearance.

### 2.2. Antibacterial Activity of the Leaf Extracts from T. dentisepala and T. lineata

In Table 1, the concentration values of the extracts from *T. dentisepala* and *T. lineata* that caused the inhibition (MIC) and death (MBC) of *E. faecalis*, *S. agalactiae*, *S. pyogenes*, *S. typhi*, *P. aeruginosa*, and *V. parahaemolyticus* bacteria can be appreciated. The hydroalcoholic leaf extract of *T. dentisepala* inhibited the growth of all evaluated bacteria in a concentration range of 1.5 to 6 mg/mL and demonstrated a bactericidal activity of 3.0 to 6.0 mg/mL. The most sensitive bacteria to the *T. dentisepala* extract were *S. typhi* and *V. parahaemolyticus*, with MIC values of 1.5 and 2.0 mg/mL, respectively, followed by *S. agalactiae* and *P. aeruginosa*, both with MIC values of 3.0 mg/mL. The *T. dentisepala* HAE exhibited *V. parahaemolyticus* activity, with an MIC of 2 mg/mL and an MBC of 4 mg/mL. Similarly, the TPEs of both species showed activities against all evaluated bacteria and required lower concentrations with respect to those of the HAEs (Table 1). On the other hand, the HAE from *T. lineata* only inhibited the growths of *S. typhi* and *P. aeruginosa*, with values of 6.0 mg/mL for the MIC and MBC, and its TPE was more active, with MIC and MBC values of 4.0 mg/mL. Both the HAE and TPE from *T. lineata* did not show antibacterial activity against the tested Gram-positive bacteria. Note that both species were collected in the same month and year.

### 2.3. ABTS^•+^ Antioxidant Activity and Total Phenolic Content 

In order to evaluate the antioxidant properties of the extracts, the ABTS^•+^-scavenging activity and concentration of total phenolic content were determined. Table 2 shows the total phenolic content (TPC) and radical-scavenging capacity of ABTS^•+^ for both hydroalcoholic extracts. The TPC in the HAE from *T. dentisepala* exhibited the highest value (64.7 ± 2.45 mg GAE/g), while the extract from *T. lineata* showed the lowest value (23.36 ± 1.26 mg GAE/g). Likewise, the extract of *T. dentisepala* demonstrated strong antiradical activity, with an IC_50_ value of 23.49 ± 1.13 µg/mL, while the extract of *T. lineata* showed lower activity by displaying an IC_50_ value of 43.72 µg/mL. Both phenolic extracts (TPEs) exhibited high antioxidant activity. The scavenging capacity of the *T. lineata* TPE exceeded the value of its HAE by 5868 times, whereas the difference between the *T. dentisepala* TPE and HAE was 3409 times. Likewise, it can be observed that the activities of the TPEs of both species were very similar.

### 2.4. High-Performance Liquid Chromatography and Automated Thin-Layer Chromatography Analyses

Figure 1 displays the HPLC profiles of the hydroalcoholic extracts from the leaves of *T. dentisepala* and *T. lineata*. These profiles were very different, which shows that both plants had their own unique chemical profiles. The *T. dentisepala* extract displayed a greater number of less polar compounds than *T. lineata* since most of the peaks presented retention times of 0–20 min, unlike *T. lineata*, where the highest intensity peaks appeared after 25–55 min, while the *T. dentisepala* extract did not exhibit these peaks. There was a peak with a retention time of 27 min, which could be a common compound present in both extracts.

The HAEs from the leaves of five individuals of *T. dentisepala* collected in the state of Nayarit and *T. lineata* from the state of Morelos, Mexico, were analyzed by thin-layer chromatography. A chromatographic mobile phase composed of chloroform and methanol (4:1) was used for the analysis of glycosylated compounds, especially for the identification of phenylethanoid glycosides. In Figure 2, the TLC profiles of the compounds observed under UV light at 365 nm (A) and 254 nm (B) revealed that the extracts from both species showed a consistent spotting pattern between the five analyzed individuals of each species. The HAE of *T. dentisepala* showed more intense blue spots at 365 nm with retention factors (Rfs) of 0.38 and 0.31 that were much less intense in the extracts from *T. lineata*, where they were practically non-existent in some individuals. In contrast, *T. lineata* showed a profile with a greater number of spots. Observing the plate under 254 nm UV light revealed the fluorescence extinction of spots with Rfs of 0.16, 0.26, and 0.32. These spots are characteristic of glycosylated phenylethanoid derivatives, including compounds previously isolated from *T. lineata*, such as ternstroside B, terngymnoside C, and hydroxytyrosol-1-glucoside [18,19]. These compounds presented a reddish color when developed with vanillin/H_2_SO_4_ and heated to 120 °C (Figure 2C). The thin-layer profile of *T. dentisepala* differed significantly from *T. lineata*, primarily due to the absence of glycosylated phenylethanoids, since no bright yellow spots were observed when the plate was developed with vanillin/H_2_SO_4_, heated to 120 °C, and observed under ultraviolet light at 365 nm (Figure 2D).

### 2.5. ^1^H Nuclear Magnetic Resonance Data

The ^1^H NMR of the hydroalcoholic extracts from the leaves of *T. dentisepala* and *T. lineata* were recorded in a 600 MHz spectrometer with the objective of knowing the chemical profiles of both plants since, unlike chromatographic methods, the analysis of NMR allows for a complete profile of the metabolites through their signals. Figure 3 displays the ^1^H NMR spectra of the two *Ternstroemia* species, with the green-colored signals corresponding to the *T. dentisepala* extract and the brown signals corresponding to the *T. lineata* species.

The ^1^H NMR analysis indicated that the extracts of *T. dentisepala* and *T. lineata* shared similar profiles with regard to the region of the spectrum corresponding to the presence of sugars (3.0–5.5 ppm), in addition to having an important content of phenolic compounds (6.0–7.5 ppm). However, the signal patterns of the phenolic compounds were different for both plants, which for *T. lineata* resembled the derived phenylethanoid glycosides, such as the previously isolated ternstroside B, which were different from those observed for *T. dentisepala.* Interestingly, *T. lineata* presented signals that corresponded to methyl groups between 0.8 and 1.3 ppm, which are characteristic for triterpenes. These signals were much less intense in the spectrum of *T. dentisepala* (Figure 3). 

## 3. Discussion

Despite the fact that close to 80% of people in Mexico and many other Latin American countries use medicinal plants to treat diseases, the development of new drugs or phytomedicines from endemic plants has been slow. The pharmaceutical industry has underappreciated the use of plant extracts to treat various health conditions, particularly after the “golden era” of antibiotic development in the 1950s. The misuse of antibiotics, which has resulted in the emergence of resistant bacterial species, in addition to the incipient development of new antibiotic classes, has fueled and escalated the current “antibacterial crisis” [20].

Acute respiratory infections were the most common type of illness in Mexico. A total of about 16 million cases of this type of ailment in the country were estimated. Urinary tract and intestinal infections were the second and third most common types of illnesses in the Mexican population, with just over three million estimated cases for each [21].

Among the 22 serogroups (49 species and 8 subspecies) of the genus *Streptococcus*, *S. pyogenes* and *S. agalactiae* are the most prominent and cause infections in humans. Different types of pharyngitis are caused by *S. pyogenes*, which is also the cause of skin infections, otitis media, and more serious infections like scarlet fever, as well as skin and mucous membrane infections [22], while *S. agalactiae* is associated with diseases caused by genital and gastrointestinal colonization [23]. *E. faecalis* is often coisolated with *P. aeruginosa* in biofilm-associated infections, such as wound infections, urinary tract infections, and periodontitis [24,25,26]. The main microbials responsible for the development of diarrhea in humans are *Escherichia coli*, *Staphylococcus aureus*, *Shigella flexneri*, and *S. typhi*, followed by *P. aeruginosa* [27,28]. Likewise, *V. parahaemolyticus* impacts human health by causing gastroenteritis; wound infections; and, in more severe cases, sepsis [29]. 

Our research showed that the hydroalcoholic leaf extract from *T. dentisepala*, particularly the total phenolic extract, was effective against all the bacteria that were tested. These findings contrast with the results obtained for *T. lineata*, which was only effective against the Gram-negative bacteria *S. typhi* and *P. aeruginosa*, which are two of the most important bacteria for finding active compounds, according to the WHO.

With MICs ranging from 1.5 to 6.0 mg/mL for all the tested bacteria, the *T. dentisepala* extract showed the highest activity. This result contrasted with the activity of the *T. lineata* extract, which only affected the *S. typhi* and *P. aeruginosa* strains, with an MIC of 6.0 mg/mL. This difference in antibacterial activity between the hydroalcoholic extracts of the two species could be explained by the presence of different secondary metabolites, such as polyphenols, tannins, flavonoids, and triterpenes, that have already been reported in other species of the *Ternstroemia* genus [16,30].

Previous reports demonstrated that the methanolic leaf extract of *T. gymnanthera* had inhibitory effects against *S. aureus* based on the induction of the largest inhibition zone of 16 mm using a paper disk with 2 mg/mL of extract [11]. In addition, it was also reported that the methanolic leaf extracts of *T. cameroonensis* present moderate activity against *E. coli*, *Klebsiella pneumoniae*, *P. aeruginosa*, *S. aureus,* and *Enterobacter aerogenes,* with MICs ranging from 0.512 to 1.024 mg/mL [30].

The *V. parahaemolyticus* bacteria was killed at 4 mg/mL and its growth was slowed down at 2 mg/mL by the hydroalcoholic extract of the leaves from *T. dentisepala*. These findings could have a positive impact on future investigations in the search for compounds to treat gastroenteritis and sepsis caused by this bacterium in humans [29]. 

Phenolic compounds comprise one of the most widely occurring groups of phytochemicals. They are also of considerable physiological and morphological importance in plants and contain secondary metabolites with antioxidant properties [10,31,32]. In the present study, we observed a relationship between the total phenolic content and the ABTS*^•+^* IC_50_ values that was higher for the HAE of *T. dentisepala* in comparison with the extract of *T. lineata*, which presented the highest total phenolic content, and lower for the ABTS*^•+^*-scavenger IC_50_ values (Table 2). Apart from the genetic differences, many other factors, including climate, soil, and height, can influence the levels of plant secondary metabolites, such as phenolic compounds [33,34]. It is well known that the antioxidant activities of plant extracts containing polyphenol components is due to their capacity to donate hydrogen atoms or electrons and scavenge free radicals [35]. The antiradical action of *T. lineata* determined in the present study (IC_50_ 43.72 µg/mL) was at the same magnitude as that reported by Salgado et al., where the methanolic extract of the leaves of *T. lineata* showed an IC_50_ value of 33.91 μg/mL in the antiradical ABTS test [18].

Furthermore, studies have reported that the total polyphenol content in the ethanolic extract of *T. sylvatica* leaves and the aqueous extracts from the flowers of *T. tepezapote* and *T. lineata* displayed the radical-scavenging capacity of 1,1-diphenyl-2-picrylhydrazyl (DPPH), with IC_50_ values of 22, 33, and 57 mg/mL, respectively [36,37]. Likewise, the antioxidant activities of other plants that are not from the *Ternstroemia* genus with similar values of the total phenolic content have been reported, such as the methanolic extracts of *Diospyros discolor* leaves [35], extracts of cultivated and wild-grown Tunisian *Ruta chalepensis* leaves [38], and methanolic extracts of the leaves from *Alstonia angustiloba* [39]. Our findings also suggest that phenolic compounds may be the major contributors to antioxidant activity, as evidenced by the antiradical activity presented by the phenolic extracts of both species.

It has been reported that plants can synthesize hydroxylated phenolic compounds, such as flavonoids, in response to microbial infection [40]. For this reason, the in vitro effectiveness of these compounds against a wide variety of microorganisms was tested and proved. Investigations revealed that phenolic compounds from natural sources exhibit potent antimicrobial activity against various clinically relevant pathogens associated with microbial infection and sensitize multi-drug resistance strains to bactericidal or bacteriostatic antibiotics [41]. Their activity is likely due to their ability to complex with extracellular and soluble proteins, as well as with bacterial cell walls, as described for quinones. More lipophilic flavonoids may also disrupt microbial membranes [10].

The TLC analyses corroborated the differences in chemical profiles between the two species (Figure 2). There was a group of metabolites that had similar retention factors (Rfs) between 0.80 and 0.95, but a less polar elution system is required to allow for chromatographic development with a better resolution of them. In Figure 2B, the TLC displays a pattern of bands with Rf 0.3–0.5 that exhibit UV light extinction at 254 nm, which is a trait of compounds previously reported in *T. lineata* [18]. These compounds correspond to phenylethanoid glycosides, such as ternstroside B, which was identified in *T. lineata* but first isolated in the Asian species *T. japonica* [13,18]. The TLC of Figure 2D shows the majority of phenylethanoid glycosides (Rf 0.3–0.5); when revealed with vanillin/H_2_SO_4_ and observed with UV light at 365 nm, they presented bright yellow colors. This pattern was not present in the TLCs of the hydroalcoholic extracts of *T. dentisepala* individuals. This observation suggests that the active compounds of *T. dentisepala*, which evidenced antibacterial and antioxidant activity, could be other groups of secondary metabolites that are not derivatives of the phenylethanoid type, but this assumption must be verified in future experiments.

According to the experiments carried out in this study, the impacts on the bacterial activity properties of the HAEs, especially the TPEs, of *T. lineata* and *T. dentisepala* could be linked to the inhibition of the growth of both Gram-positive and Gram-negative bacteria by the presence of phenolic compounds. The higher concentration of phenolic compounds in *T. dentisepala* or the presence of more active phenolic compounds compared with *T. lineata* may have been the cause of the observed increase in antibacterial activity in the TPE compared with the HAE of each species. Future work must address these hypotheses by isolating individual phenolic compounds and determining their antibacterial activities.

## 4. Materials and Methods

### 4.1. Plant Material Collection

Leaves from both species were collected in September 2023 in two different states of Mexico: Tepic in Nayarit State (coordinates 21,486795, −104.981792) and Huitzilac in Morelos State (coordinates 19.026338, −99.274231). The plant material was identified by M. Sc. Gabriel Flores Franco, the curator of the HUMO herbarium at the Morelos Authonomous Statal University (Morelos, Mexico), where the voucher specimen numbers (28,938 and 28,941) were deposited.

### 4.2. Plant Extracts Preparation

The plant materials were dried in a cool, dry place without direct sunlight, and then each was milled to obtain a powder. After that, they were mixed with distilled water and methanol (J. T. Baker, Radnor, PA, USA) in a 1:1 proportion in a 1:3 ratio (g/mL solvent), sonicated for 30 min, and then filtered (0.45 μm, HPF-Nylon Millex^®^, Merck, Darmstadt, Germany) in triplicate. To obtain the hydroalcoholic crude extracts, the filtered liquids were evaporated under vacuum (Rotavapor R-124, Büchi, Flawil, Switzerland) at 40 °C until dryness.

### 4.3. Total Phenolic Extract Preparation

In order to obtain a total phenolic extract, an Amberlite^™^ XAD^®^-2 resin (Merck, Darmstadt, Germany) was used in an open column (15 × 200 mm). Each HAE (200 mg) was dissolved in 20 mL of acidified water (pH 2.0) before applying it to the column. The washing step entailed adding 200 mL of water to elute all non-phenolic compounds. Then, the resin was eluted with 100 mL of ethanol (J. T. Baker, Radnor, PA, USA) to obtain the total phenolic extract, and the solvent was evaporated in a rotavapor under reduced pressure at 40 °C.

### 4.4. Growth Conditions of Bacterial Strains and Culture

All microbiological media were purchased from Sigma-Aldrich (St. Louis, MO, USA). A pure bacterial culture of the Gram-positive *Enterococcus faecalis* (ATCC 29212), *Streptococcus agalactiae* (ATCC 25924), and *Streptococcus pyogenes* (ATCC 19615), along with the Gram-negative *Salmonella typhi* (ATCC 6539), *Pseudomonas aeruginosa* (ATCC 9027), and *Vibrio parahaemolyticus* (Vp417 strain), were obtained by inoculation in Mueller–Hinton broth at 37 °C with shaking (150 rpm) overnight. The microorganisms were preserved at 4 °C and sub-cultured at regular intervals of 30 days. All other chemicals and reagents were of the highest analytical grade and commercially available. The Vp417 strain was isolated from the hepatopancreas of a dying shrimp with signs of acute hepatopancreatic necrosis disease that was captured on a farm in San Blas Nayarit State, Mexico, in 2014. Bacterial identification was carried out using 16S primers, including a control strain. The BLAST analysis revealed that the control strain’s sequence had 99% identity and 100% coverage with the nucleotide sequence of *Vibrio parahaemolyticus* compared with the deposited sequence in GenBank.

### 4.5. Antibacterial Activity of T. dentisepala and T. lineata Extracts

The antibacterial assay of the hydroalcoholic and total phenolic extracts of both populations was performed by the microdilution technique using 96-well plates that allowed for the determination of the minimum inhibitory concentration (MIC) [42]. Isolated colonies were used to prepare bacterial suspensions in 5 mL of Cation-Adjusted Mueller-Hinton Broth (CAMHB, BD^™^, Le Pont de Claix, France) and then incubated until reaching a 0.5 McFarland standard. The bacterial suspensions were diluted to achieve a final inoculum of 5 × 10^5^ CFU in each well. The stock solutions of the extracts were serially diluted 2-fold in water to final concentrations between 0.75 and 6 mg/mL. The extracts presented a red/yellow coloration. For this reason, an extract control (medium without inoculum) was included, and also a sterility check (medium and water), a negative control (medium and inoculum), and a positive control (medium, inoculum, and the appropriate antibiotics: gentamicin at 4 μg/mL, chloramphenicol at 8 μg/mL, or ampicillin at 8 μg/mL) were considered for each experiment. After 18 h of incubation at 37 °C, 10 μL of resazurin (22 μM) (Merck, Darmstadt, Germany) was added and incubated for two more hours. The MIC of each extract was established as the lowest concentration that did not turn pink. To determine the minimum bactericidal concentration (MBC), prior to the addition of resazurin, an aliquot was taken from the wells where the turbidity was absent and then inoculated in a Petri dish with Mueller–Hinton agar (BD^™^, Le Pont de Claix, France). The concentration point at which the bacterial growth was not observed was defined as the MBC value.

### 4.6. Antiradical ABTS^•+^ Assay

The scavenging activity of hydroalcoholic and total phenolic extracts was evaluated with the ABTS method [18]. A stock of ABTS radical cation (ABTS^•+^) (Merck, St. Louis, MO, USA) was prepared by the reaction of 7 mM ABTS^•+^ solution with 245 mM of ammonium persulfate and incubated for 16 h in the dark. The ABTS^•+^ solution was diluted with ultrapure water to obtain an absorbance of 0.7 (±0.1) at 734 nm. Ten microliters of different concentrations of the extracts were added to a 96-well plate and mixed with the ABTS^•+^ solution (190 μL). After 5 min of incubation in the dark, the absorbance was measured at 734 nm. Quercetin (Sigma-Aldrich, St. Louis, MO, USA) was used as a positive control. The antiradical activity was expressed as IC_50_ (μg/mL), which represents the extract concentration that scavenged 50% of ABTS radicals.

### 4.7. Determination of the Total Phenolic Content 

The total phenolic content of individual extracts was determined using the Folin–Ciocalteu method [43]. Briefly, 100 μL of extract (500–1000 μg/mL) solution was mixed with 400 μL of ultrapure water and 150 μL of 50% (*w*/*v*) Folin–Ciocalteu reagent (HYCEL, Mexico City, Mexico). After 5 min, 500 μL of 20% Na_2_CO_3_ was subsequently added to the mixture and incubated at 50 °C for 1 h in the dark. The sample absorbance was measured using a microplate reader (SpectraMax^®^ iD3 Plus 384, Molecular Devices, San Jose, CA, USA) at 710 nm and was compared with a blank without an extract. The outcome data were expressed as mg/g of gallic acid equivalents in milligrams per gram (mg GAE/g) of dry extract. 

### 4.8. High-Performance Liquid Chromatography Analysis

The HPLC analysis was used to obtain the chromatographic profile of the leaf extracts from both species of *Ternstroemia*. The chromatographic analysis was carried out on a Jasco AS-4150 liquid chromatograph, which included a Jasco RHPLC Autosampler AS-4150 injector, a Jasco RHPLC Pump PU-4180 quaternary pump, and a Jasco UV-4075 UV/Visible detector (Tokyo, Japan). A stationary phase of silica C-18 with 5 μm in a 4.6 mm × 250 mm (Waters, Milford, MA, USA) and a mobile phase of acetonitrile and water (85:15) with a flow of 1 mL/min were used. A solution of the extract to be analyzed was prepared at a concentration of 5 mg/mL in methanol. An aliquot of 10 μL was used as the injection volume. The ChromNAV 2.0 program performed peak recording. A wavelength of 254 nm was used for the detection. The solvents used for the chromatographic processes were of HPLC grade (J.T. Baker, Radnor, PA, USA).

### 4.9. Automated Thin-Layer Chromatography Analysis

Ten milligrams of each extract were resuspended in 1 mL of methanol, and then 3 μL were applied to the TLC plate using precoated aluminum sheets of silica gel 60 UV_254,_ with a particle size of 5 μm, and then developed in the CAMAG Linomat 5 instrument (Muttenz, Switzerland). Derivatization was carried out by spraying vanillin solution (1 g vanillin in 100 mL of H_2_SO_4_) and heating at 120 °C until the color plate changed. Visualization and analysis of the plates were performed with visible light and UV light at 254 and 365 nm as necessary using the VISION CATS program. The mobile phase was composed of chloroform and methanol (4:1) of analytical grade (J.T. Baker, Radnor, PA, USA).

### 4.10. ^1^H Nuclear Magnetic Resonance Analysis

The ^1^H NMR experiments were performed on a Jeol ECZ 600 MHz (Peabody, MA, USA) with a Royal NMR Proton Optimized High Sensitivity 5 mm Probe. The hydroalcoholic extract (5 mg) was solubilized in 700 μL of CD_3_OD.

## 5. Conclusions

Although further clinical studies are required, the hydroalcoholic extracts, especially the total phenolic extract of *T. dentisepala*, contain active compounds against bacteria that may be able to control diseases caused by *E. faecalis*, *S. agalactiae*, *S. pyogenes*, *S. typhi*, *P. aeruginosa*, and *V. parahaemolyticus*. This research offers new insights into the bacterial control of clinical interests.

## Figures and Tables

**Figure 1 plants-13-02515-f001:**
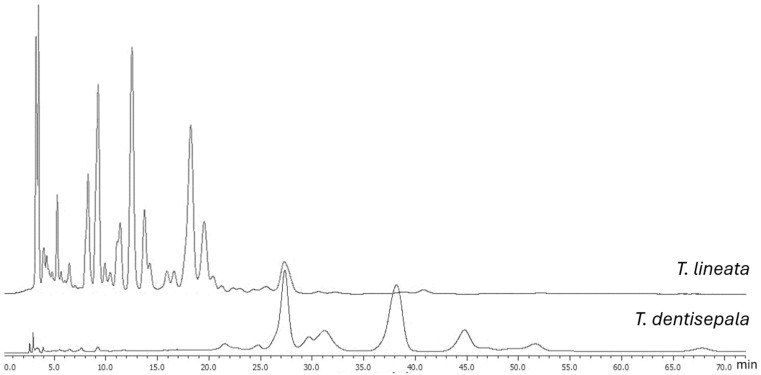
HPLC analysis of the hydroalcoholic extracts from the leaves of *T. lineata* and *T. dentisepala*. Chromatographic conditions: C-18 column and a mobile phase of acetonitrile and water (85:15) with a flow of 1 mL/min and detection at 254 nm.

**Figure 2 plants-13-02515-f002:**
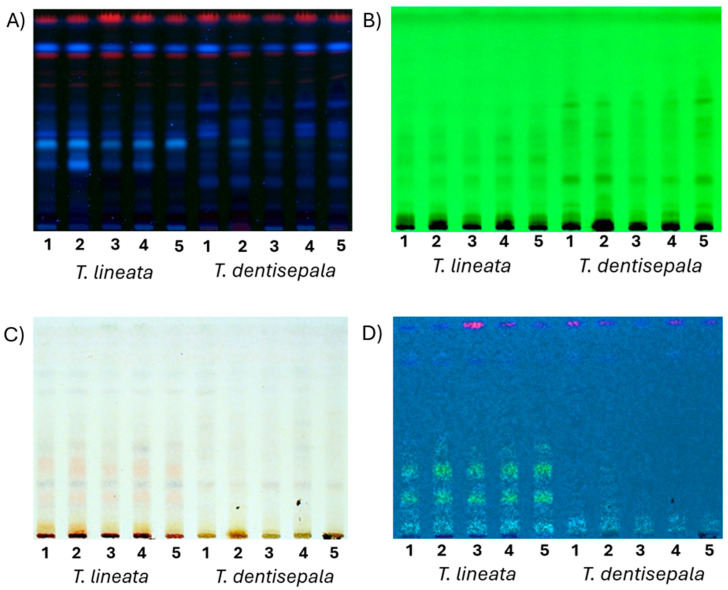
Thin-layer chromatographic profiles of the hydroalcoholic extracts of five individuals of *T. dentisepala* and *T. lineata*. The numbers 1, 2, 3, 4, and 5 represent extracts from different individuals of the same species. (**A**) UV light at 365 nm; (**B**) UV light at 254 nm; (**C**) plate developed with vanillin/H_2_SO_4_ and 120 °C; (**D**) plate developed with vanillin/H_2_SO_4_ and observed with UV light at 365 nm. The elution system was composed of a mobile phase with chloroform and methanol (4:1) in silica gel.

**Figure 3 plants-13-02515-f003:**
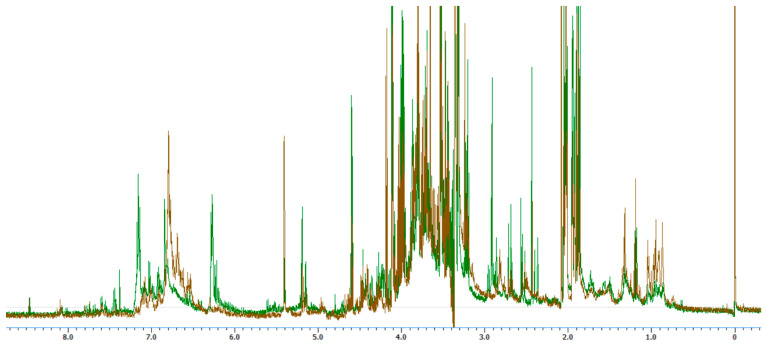
^1^H NMR of the hydroalcoholic extracts from the leaves of *T. dentisepala*, in green color, and *T. lineata*, in brown color, at 600 MHz in a CD_3_OD solvent.

**Table 1 plants-13-02515-t001:** Minimum inhibitory concentration (MIC) and minimum bactericidal concentration (MBC) in mg/mL of the hydroalcoholic (HAEs) and the total phenolic extracts (TPEs) of *T. dentisepala* and *T. lineata*.

Extracts and Controls	Gram-Positive Bacteria						Gram-Negative Bacteria					
	*Enterococcus faecalis*		*Streptococcus agalactiae*		*Streptococcus pyogenes*		*Salmonella typhi*		*Pseudomonas aeruginosa*		*Vibrio parahaemolyticus*	
	MIC	MBC	MIC	MBC	MIC	MBC	MIC	MBC	MIC	MBC	MIC	MBC
*T. dentisepala* (HAE)	6.0	6.0	3.0	3.0	6.0	6.0	1.5	1.5	3.0	3.0	2.0	4.0
*T. dentisepala* (TPE)	4.0	4.0	2.0	2.0	4.0	4.0	1.0	1.0	2.0	2.0	1.0	2.0
*T. lineata * (HAE)	nd	-	nd	-	nd	-	6.0	6.0	6.0	6.0	nd	-
*T. lineata * (TPE)	nd	-	nd	-	nd	-	4.0	4.0	4.0	4.0	nd	-
Positive controls												
Gentamicin	-	-		-	-	-	0.004	-	0.004	-	0.004	-
Chloramphenicol	0.008	-	0.008	-	0.008	-	0.008	-	-	-	0.008	-
Ampicillin	0.008	-	0.008	-	0.008	-	-	-	0.008	-	-	-

nd—not detected.

**Table 2 plants-13-02515-t002:** *ABTS^•+^*-scavenging activity of the hydroalcoholic extracts (HAEs) and the total phenolic extract (TPE), and the total phenolic content in the HAEs from *T. dentisepala* and *T. lineata*.

Plant Extract	ABTS^•+^-Scavenging IC_50_ µg/mL	Total Phenolic Content (mg GAE/g DW)
*T. dentisepala* HAE	23.49 ± 1.1 ^b^	64.7 ± 2.4 ^B^
*T. dentisepala* TPE	16.44 ± 1.8 ^c^	nd
*T. lineata* HAE	43.72 ± 1.0 ^e^	23.36 ± 1.3 ^A^
*T. lineata* TPE	32.79 ± 2.1 ^d^	nd
Quercetin	11.2 ± 0.8 ^a^	nd

Data represent mean ± standard deviation from testing performed in triplicate; DW: dry weight; GAE: gallic acid equivalent; nd—not determined. Similar letters show no significant statistical difference according to the Tukey test (*p <* 0.05).

## Data Availability

The data presented in this study are available upon request from the corresponding authors.

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
