# Peer review of "Antibacterial and Antioxidant Activities of Hydroalcoholic and Phenolic Extracts from *Ternstroemia dentisepala* and *T. lineata* Leaves"

_plants, 2024, doi:10.3390/plants13172515_

Round 1
Reviewer 1 Report
Comments and Suggestions for Authors
It is necessary to write Salmonella Typhi resp. S. Typhi everywhere in the text of the article instead Salmonella typhi, since it is a serovariant of Salmonella enterica. Salmonella Typhi is an the accepted abbreviated spelling, but with a capital letter of the name of the serovariant, instead the full serovar name: Salmonella enterica ssp. enterica serovariant typhi.
The family name "Enterobacteriaceae" on lines 64-65 of the introduction should be in italics. Ibid., Salmonella spp. is not written correctly, because now the Salmonella species is one - Salmonella enterica, with 6 subspecies and many serovars. That's why instead of “Salmonella spp.”, it should be “Salmonella enterica”.
In the Results section and below, the names of the bacterial species should be abbreviated, as they have already been given in full above. As a rule, after their initial complete writing, they are further written abbreviated.
The sentence on lines 91-92 is a discussion and should not present in the Results section. As there are discussion items elsewhere in this section (lines 150-151) I suggest that they be moved in "Discussion" or that the section be titled "Results and Discussion".
On line 185, I suggest that "resistant species" should become "resistant bacterial species".
The text on lines 207-212 is a description of the results and should be in the preceding section, not in the Discussion. This also applies to the text of lines 227-235.
On line 248, I suggest that "the filtered" should become "the filtered liquids".
There is no information on the origin of the MIC plates used in the Materials and Methods section. It is appropriate to add the manufacturing companies of the consumables used, as well as the equipment (for sonification of the extracts, 0.45 μm filters, etc.).
In the conclusion, the text of lines 313-316 should be revised, as it is not a direct result of the research carried out. The text "These species hold significant medicinal value in Asia and Mexico, where they are integral to traditional medicine" should be dropped from the conclusions. Clinical studies proving "the potential to control diseases caused by Enterococcus faecalis, Streptococcus agalactiae, Streptococcus pyogenes, Salmonella typhi, Pseudomonas aeruginosa, and Vibrio parahaemolyticus..." have not been carried out. Instead, it should be written that antibacterial activity in vitro of the tested extracts against the examined bacteria has been demonstrated.
References 1 and 7 are not complete.
Author Response
It is necessary to write Salmonella Typhi resp. S. Typhi everywhere in the text of the article instead Salmonella typhi, since it is a serovariant of Salmonella enterica. Salmonella Typhi is an the accepted abbreviated spelling, but with a capital letter of the name of the serovariant, instead the full serovar name: Salmonella enterica ssp. enterica serovariant typhi.
Answer: We appreciate your feedback, which contributed to the manuscript's improvement. We heeded your observation and used the name "Salmonella Typhi" throughout the text.
The family name "Enterobacteriaceae" on lines 64-65 of the introduction should be in italics. Ibid., Salmonella spp. is not written correctly, because now the Salmonella species is one - Salmonella enterica, with 6 subspecies and many serovars. That's why instead of “Salmonella spp.”, it should be “Salmonella enterica”.
Answer: The observations were taken care of.
In the Results section and below, the names of the bacterial species should be abbreviated, as they have already been given in full above. As a rule, after their initial complete writing, they are further written abbreviated.
Answer: We agree with this and have incorporated your suggestion throughout the manuscript.
The sentence on lines 91-92 is a discussion and should not present in the Results section. As there are discussion items elsewhere in this section (lines 150-151) I suggest that they be moved in "Discussion" or that the section be titled "Results and Discussion".
Answer: We removed the information from the results section and referenced it in the discussion section.
On line 185, I suggest that "resistant species" should become "resistant bacterial species".
Answer: It was modified.
The text on lines 207-212 is a description of the results and should be in the preceding section, not in the Discussion. This also applies to the text of lines 227-235.
Answer: We agree with this and moved the sentences to the results section.
On line 248, I suggest that "the filtered" should become "the filtered liquids".
Answer: It was modified.
There is no information on the origin of the MIC plates used in the Materials and Methods section. It is appropriate to add the manufacturing companies of the consumables used, as well as the equipment (for sonification of the extracts, 0.45 μm filters, etc.).
Answer: Information about manufacturing companies, as well as the state and country of origin of equipment, materials, and consumables, was completed in the materials and methods section.
In the conclusion, the text of lines 313-316 should be revised, as it is not a direct result of the research carried out. The text "These species hold significant medicinal value in Asia and Mexico, where they are integral to traditional medicine" should be dropped from the conclusions. Clinical studies proving "the potential to control diseases caused by Enterococcus faecalis, Streptococcus agalactiae, Streptococcus pyogenes, Salmonella typhi, Pseudomonas aeruginosa, and Vibrio parahaemolyticus..." have not been carried out. Instead, it should be written that antibacterial activity in vitro of the tested extracts against the examined bacteria has been demonstrated.
Answer: We agree with your comments. As a result, we corrected and updated the conclusions to be more accurate and improved.
References 1 and 7 are not complete.
Answer: We completed the references.

Reviewer 2 Report
Comments and Suggestions for Authors
The authors investigated hydroalcoholic extracts from the leaves of Tersntroemia dentisepala and T. lineata using microdilution methodology against bacteria involved in human health, including Enterococcus faecalis, Streptococcus agalactiae, Streptococcus pyogenes, Salmonella typhi, Pseudomonas aeruginosa and Vibrio parahaemolyticus. The T. dentisepala extract was active against all bacterial strains tested, while T. lineata extract only demonstrated efficacy against S. typhi and P. aeruginosa. However, levels of antibacterial activity required high concentrations of extracts. Chemical analyzes by HPLC and proton nuclear magnetic resonance (1H NMR) confirm that both extracts have different chemical profiles, in the content of phenolic compounds.
Author Response
Reviewer 2 did not make any request for correction of the manuscript.
Reviewer 3 Report
Comments and Suggestions for Authors
The manuscript titled “Antibacterial activity of the hydroalcoholic extracts from Ternstroemia dentisepala and T. lineata leaves” investigates the traditional Mexican medicine(Ternstroemia species) which are commonly used to treat insomnia, injuries and infections for its antibacterial activity. The authors have reported the antibacterial activity of hydroalcoholic extracts from the leaves of Ternstroemia species against a panel of Gram-positive and Gram-negative bacteria for the first time. Despite the novelty and the potential economic value of the work ,the manuscript requires more investigations and several more experiments are needed. Therefore, the current state manuscript is not suitable to publication in “Plants” . Please find the comments below.
Comments:
1. Chemical analysis of the crude extract is carried out using TLC, HPLC, and 1H NMR. However, testing the crude extract alone may yield inaccurate results due to potential synergistic or antagonistic effects of individual compounds, leading to biased conclusions. To accurately screen for natural antibiotic compounds, it is recommended to fractionate the crude extract and conduct chemical profiling using these techniques for more precise identification.
2. As the authors have mentioned about difference about chemical profiles in terms of phenolic compounds, total phenolic content in both extracts should be evaluated.
3. “…Ternstroemia spp. represent a potential source of new antibacterials medicines since many of their ethnomedical uses are related to problems of bacterial infection, and their research could result in the development of new drugs”. Please provide appropriate references for this statement.
Comments on the Quality of English LanguageModerate editing of the English language is required.
Author Response
Comments:
- Chemical analysis of the crude extract is carried out using TLC, HPLC, and 1H NMR. However, testing the crude extract alone may yield inaccurate results due to potential synergistic or antagonistic effects of individual compounds, leading to biased conclusions. To accurately screen for natural antibiotic compounds, it is recommended to fractionate the crude extract and conduct chemical profiling using these techniques for more precise identification.
Answer: The use of plant extracts in Mexico, as well as in various other countries, is part of a deep-rooted practice in traditional medicine. The determination of the pharmacological activities of plant extracts has significant scientific value, as does the rational exploitation of local natural resources. Furthermore, the knowledge of the antibacterial activity of plant extracts, as in the case of the present investigation, where two plants of the same genus are analyzed, which in the case of the Ternstroemia species that occur in Mexico most of the time are used indiscriminately under the name of a plant material called "tila" contributes to promoting the need for strict quality control. The current study illustrates that the chemical profiles of both Ternstroemia species differ, leading to distinct antibacterial properties. Therefore, the application of the "tila" plant material necessitates accurate botanical identification, irrespective of its intended uses as domestic teas or as raw material for large-scale production and marketing.
This project offers new insights into the phytochemical and pharmacological properties of the "tila" species that are used and marketed in Mexico. It requires further chemical characterization of the individual compounds of each species, a process that could take several months or years. We have made significant efforts to gather the required reagents from the Amberlite XAD-2 resin to extract a rich fraction of phenolic compounds from Ternstroemia lineata and T. dentisepala. We incorporated the results of these phenolic extracts' antibacterial activity into the updated version of the manuscript, finding them to be more active than the crude hydroalcoholic extract.
At the moment, my research group has two new postgraduate students who will begin a research project to purify the major phenolic compounds of both plants. The purified compounds will aid in enhancing the characterization of individual constituents and their pharmacological activities, including antibacterial properties. We recently submitted an article that was accepted for publication, which details the antioxidant activity in silico, in vitro, and in vivo of three compounds isolated from Tersntroemia lineata. This information has been updated in the current manuscript. However, due to the limited amount of the three isolated compounds, we have not tested their antibacterial activity, necessitating further isolation.
- As the authors have mentioned about difference about chemical profiles in terms of phenolic compounds, total phenolic content in both extracts should be evaluated.
Answer: The total phenolic content was evaluated as requested by the reviewer. In addition, total phenolic extracts were obtained from the use of Ambertil XAD-2 resin, which were evaluated and showed to be more active than hydroalcoholic extracts in the antibacterial test. We also determined the extracts' antiradical activity using the ABTS method.
- “…Ternstroemia spp. represent a potential source of new antibacterials medicines since many of their ethnomedical uses are related to problems of bacterial infection, and their research could result in the development of new drugs”. Please provide appropriate references for this statement.
Answer: The article by Villanueva-Solis et al. (2020) describes the use of tila for treating nerves, fear, insomnia, and infections. The use cases for infections refer primarily to treating diarrhea and dysentery, as well as sore throats. Furthermore, Juárez-Pérez and Cabrera-Luna (2019) report the use of tila for the treatment of cough, while Fernández-Nava et al. (2001) describe the plant's use for treating respiratory tract infections.
Villanueva-Solis, I.; Arreguín-Sánchez, M.L.; Quiroz-García, D.L.; Fernández-Nava, R.F. Medicinal plants sold in the 8 July market and a traditional market, both located in the center of Actopan, Hidalgo, Mexico. Polibotánica. 2020, 50, 209-243. https://doi.org/10.18387/polibotanica.50.14
Fernández-Nava, R.F.; Ramos-Zamora, D.; Carranza-González, E. Notas sobre las plantas medicinales del estado de Querétaro, México. Polibotánica. 2001, 12, 1-40. https://polibotanica.mx/index.php/polibotanica/article/view/662
Juárez-Pérez, J.C.; Cabrera-Luna, J.A. Plants to treat respiratory ailments that are commercialized in three markets in Querétaro city. Polibotánica. 2019, 47, 167-178. https://doi.org/10.18387/polibotanica.47.12

Reviewer 4 Report
Comments and Suggestions for Authors
"Antibacterial activity of the hydroalcoholic extracts from 2 Ternstroemia dentisepala and T. lineata leaves" (Manuscript ID plants-3096275). I appreciate the authors' effort in conducting this study. The manuscript addresses an important topic and contains valuable information. However, there are several areas that require revision to improve the clarity, coherence, and overall quality of the paper. Below are my detailed comments and suggestions for the authors:
The research are presented the antibacterial effects of two species of plants from Ternstroemia spp. following bacterial strain culture but the results are not clearly stated, there is only a table presented briefly their effects following the different concentrations but the results parts did not detailed well that results but it was found in the discussion, leaving less information to justify those results. it was the same with the determination of secondary metabolites found in those species, interpret in the discussion but at the end if still difficult to really identify what was those secondary metabolites involved in the different bacterial strains.
I think the authors should process more experiments to clearly determine and analyses those components involved in the antibacterial effects to make the work complete.
Comments on the Quality of English Language
English editing language is quite well-written and just need minor revisions
Author Response
"Antibacterial activity of the hydroalcoholic extracts from 2 Ternstroemia dentisepala and T. lineata leaves" (Manuscript ID plants-3096275). I appreciate the authors' effort in conducting this study. The manuscript addresses an important topic and contains valuable information. However, there are several areas that require revision to improve the clarity, coherence, and overall quality of the paper. Below are my detailed comments and suggestions for the authors:
The research are presented the antibacterial effects of two species of plants from Ternstroemia spp. following bacterial strain culture but the results are not clearly stated, there is only a table presented briefly their effects following the different concentrations but the results parts did not detailed well that results but it was found in the discussion, leaving less information to justify those results. it was the same with the determination of secondary metabolites found in those species, interpret in the discussion but at the end if still difficult to really identify what was those secondary metabolites involved in the different bacterial strains.
I think the authors should process more experiments to clearly determine and analyses those components involved in the antibacterial effects to make the work complete.
Answer: We valued and considered the reviewer's pertinent observations. We expanded the presentation of the results to include both the antibacterial activity and the extraction process. In addition, the determination of total phenols by the Folin-Ciocalteu method was included and the total phenolic extracts of the two Ternstroemia species were obtained, which were evaluated in the antibacterial model. We also determined the extracts' antiradical activity using the ABTS method.
Comments on the Quality of English Language
English editing language is quite well-written and just need minor revisions
Answer: We reviewed the entire text in terms of the English language and editing format.
Reviewer 5 Report
Comments and Suggestions for Authors
The article with the title "Antibacterial activity of the hydroalcholic extracts from Ternstroemia dentisepala and T. lineata leaves", preliminary results regarding the compounds contained in the species. On the chemistry side, the article is poor in results, there is no compound identified in the extracts analysed by any technique presented.
Suggestions for improvement:
1. In the introduction part, there is no scientific article regarding the chemical composition of this plant. Haven't studies been done so far by other researchers?
2. At chapter 2, results, please explain what means 1, 2, 3, 4, 5 in figure 2. Also on figure 1, HPLC chromatogram there wasn't explication about her. Each chromatographic peak should correspond to a compound. What would they be I do not think that the data provided are sufficient for an article in this journal. What would be the compounds responsible for antimicrobial activity?
Certainly, extracts also have antioxidant potential that should be exploited together with the antimicrobial one.
3. In the methodology part, the concentration of the ethanol extracts from which it is based is not presented, nor what amount of extract is injected into HPLC.
The information obtained is not correlated with each other and the data presented are poor. They seem like information obtained in a first experiment. They should be resumed and deepened.
Author Response
The article with the title "Antibacterial activity of the hydroalcholic extracts from Ternstroemia dentisepala and T. lineata leaves", preliminary results regarding the compounds contained in the species. On the chemistry side, the article is poor in results, there is no compound identified in the extracts analysed by any technique presented.
Suggestions for improvement:
In the introduction part, there is no scientific article regarding the chemical composition of this plant. Haven't studies been done so far by other researchers?
Answer: We thank the reviewer for the pertinent comments. In the introduction, information was added regarding the compounds previously isolated and identified from Ternstroemia lineata (before T. pringlei), which were published by our research group, one in 2015 and another more recently accepted for publication (July 2024).
More information was added about determination of the total phenolic content using the Folin-Ciocalteu method in order to learn more about the antibacterial activity observed in the extracts of T. lineata and T. dentisepala. Additionally, we obtained the total phenolic fractions of both plants by using Amberlite XAD-2 resin. Both phenolic extracts were more active than the hydroalcoholic extracts in the antibacterial model. The updated version of the manuscript also includes the extracts' antiradical activity.
At chapter 2, results, please explain what means 1, 2, 3, 4, 5 in figure 2. Also on figure 1, HPLC chromatogram there wasn't explication about her. Each chromatographic peak should correspond to a compound. What would they be I do not think that the data provided are sufficient for an article in this journal. What would be the compounds responsible for antimicrobial activity?
Answer: The numbers 1, 2, 3, 4, and 5 in figure 2 represent extracts from different individuals of the same species. The idea of viewing an individual's profile is to observe variations that may occur in their chemical profiles. The information was clarified in the figure´s caption.
At this point in the research, the goal of presenting an HPLC profile was to demonstrate the significant difference that exists between the hydroalcoholic extracts of the two plants analyzed, which are very different. T. lineata has a significant content of more polar compounds, whereas T. dentisepala has a predominantly less polar composition. In 2015, we published an article on the chemical composition of T. lineata (formerly T. pringlei), in which we identified the presence of the glisosylated phenylethanoid Ternstroside B in the plant's calyx. However, we have not identified this compound in the plant's leaves or raw flowers more recently. We conclude that T. lineata has a diverse chemical composition depending on the organ analyzed, in addition to the fact that it may vary depending on the time of collection.
The objective of the present research, which is submitted for publication in the special edition on "bioactive extracts," is to delve into antibacterial activity studies that are nonexistent for the Ternstroemia species that occur in Mexico. The aim of this work is to determine which plant has a greater antibacterial potential, so we will continue to know the individual compounds, which indicate that they are of the phenolic type, with antibacterial activity.
Certainly, extracts also have antioxidant potential that should be exploited together with the antimicrobial one.
Answer: Yes, we agree with the reviewer. We also determined the antiradical capacity of the extracts using the ABTS method, which enriched the discussion of the manuscript.
- In the methodology part, the concentration of the ethanol extracts from which it is based is not presented, nor what amount of extract is injected into HPLC.
Answer: This part was completed in the materials and methods section.
The information obtained is not correlated with each other and the data presented are poor. They seem like information obtained in a first experiment. They should be resumed and deepened.
Answer: As previously mentioned, we made great efforts to improve the manuscript in terms of experiments, results, and discussion. Even though it's the holiday season, we were able to buy the special chemicals we needed to determine the total phenolic content in the hydroalcoholic extract of both plants, get the total phenolic extract by using Amberlit XAD-2 resin, test the extracts for antibacterial activity, and test the extracts for antiradical activity using the ABTS method.
We believe that this manuscript provides new chemical and pharmacological information on two Mexican plants widely used in traditional medicine, as well as for the production of teas marketed in mainstream supermarkets in the country. This manuscript opens new perspectives for work on the antibacterial activity of extracts from T. lineata and T. dentisepala. Since there are no scientific reports available for the latter plants, this manuscript is highly relevant to the study of these medicinal plants in Mexico. This work establishes the groundwork for future studies that will deepen our understanding of each plant's individual constituents and the pharmacological activities of these compounds.
Round 2
Reviewer 4 Report
Comments and Suggestions for Authors
The Manuscript, ID plants-3096275, entitled "Antibacterial activity of the hydroalcoholic extracts from Ternstroemia dentisepala and T. lineata leaves" was quite well revised by the authors helping to understand better their study, improving the quality of the paper. However, there remain minor modifications:
- The structure of the abstract is confusing, the authors started by presenting a part of the work's objectives and then added a brief introduction (Line 15-16). I suggest reversing the order to make it quite clear. At the same time, the authors used the impersonal form with "the antibacterial activity..... was evaluated" and the personal form, such as "we evaluated.....", I suggest choosing only one form applied in the whole manuscript (the impersonal form is usually more convenient).
- Material and methods: there are a few grammatical mistakes that have to be checked. e.g. Line 385: "diluted twofold..." should be "diluted 2-fold..."; Line 436-437: "Visualization and analysis of the plates will be performed...." should be "Visualization and analysis of the plates were performed..."; and more others errors. Line 388-389: "A positive control (medium, inoculum and respective antibiotics), Which antibiotics are you talking about? I found them in the results (Table 1) but it was not mentioned anywhere in the manuscript. I suggest mentioning them here. Line 398: I recommend adding the significance of the abbreviation ABTS when it is used for the first time in the manuscript.
- The conclusion ended by mentioning the new perspectives of these plants on bacterial control in shrimp farming, but nowhere in the manuscript it was mentioned this interest except for the isolation of bacteria strains in material and methods and the discussion, but not enough to be stated as the last sentence of the manuscript.
Author Response
REVISOR 1:
Comments and Suggestions for Authors
The Manuscript, ID plants-3096275, entitled "Antibacterial activity of the hydroalcoholic extracts from Ternstroemia dentisepala and T. lineata leaves" was quite well revised by the authors helping to understand better their study, improving the quality of the paper. However, there remain minor modifications:
- The structure of the abstract is confusing, the authors started by presenting a part of the work's objectives and then added a brief introduction (Line 15-16). I suggest reversing the order to make it quite clear. At the same time, the authors used the impersonal form with "the antibacterial activity..... was evaluated" and the personal form, such as "we evaluated.....", I suggest choosing only one form applied in the whole manuscript (the impersonal form is usually more convenient).
ANSWERS: Many thanks for your insightful comments.
We revised the abstract and considered the recommendations.
We have marked the modifications in the manuscript in blue.
We revised the entire text to enable its impersonal use, primarily in the methodology description.
-Material and methods: there are a few grammatical mistakes that have to be checked. e.g. Line 385: "diluted twofold..." should be "diluted 2-fold..."; Line 436-437: "Visualization and analysis of the plates will be performed...." should be "Visualization and analysis of the plates were performed..."; and more others errors. Line 388-389: "A positive control (medium, inoculum and respective antibiotics), Which antibiotics are you talking about? I found them in the results (Table 1) but it was not mentioned anywhere in the manuscript. I suggest mentioning them here. Line 398: I recommend adding the significance of the abbreviation ABTS when it is used for the first time in the manuscript.
ANSWERS:
It was corrected to “2-fold” (line 383).
The antibiotics and their concentrations were described: “appropriate antibiotics: gentamicin at 4 ug/mL, chloramphenicol at 8 ug/mL, or ampicillin at 8 ug/mL)” (lines 387-388).
It was corrected to “Visualization and analysis of the plates were performed” (line 436).
The significances of the abbreviation ABTS and also of the DPPH were added to the text when they appeared for the first time (lines 20 and 291, respectively).
We found some additional typos, corrected them, and marked them in blue in the text.
-The conclusion ended by mentioning the new perspectives of these plants on bacterial control in shrimp farming, but nowhere in the manuscript it was mentioned this interest except for the isolation of bacteria strains in material and methods and the discussion, but not enough to be stated as the last sentence of the manuscript.
ANSWER: Thank you for your comment. We have corrected this issue in the text.

Reviewer 5 Report
Comments and Suggestions for Authors
The authors have significantly improved the article taking into account all the suggestions made. I recommend publishing it in an updated form.
Author Response
REVIEWER 2:
Comments and Suggestions for Authors
The authors have significantly improved the article taking into account all the suggestions made. I recommend publishing it in an updated form.
ANSWER: We appreciate your review and input on this manuscript.
